# A second DNA binding site in human BRCA2 promotes homologous recombination

Catharina von Nicolai[1], Åsa Ehlén[1], Charlotte Martin[1], Xiaodong Zhang[2] & Aura Carreira[1]

BRCA2 tumour-suppressor protein is well known for its role in DNA repair by homologous recombination (HR); assisting the loading of RAD51 recombinase at DNA double-strand breaks. This function is executed by the C-terminal DNA binding domain (CTD) which binds single-stranded (ss)DNA, and the BRC repeats, which bind RAD51 and modulate its assembly onto ssDNA. Paradoxically, analysis of cells resistant to DNA damaging agents missing the CTD restore HR proficiency, suggesting another domain may take over its function. Here, we identify a region in the N terminus of BRCA2 that exhibits DNA binding activity (NTD) and provide evidence for NTD promoting RAD51-mediated HR. A missense variant detected in breast cancer patients located in the NTD impairs HR stimulation on dsDNA/ssDNA junction containing substrates. These findings shed light on the function of the N terminus of BRCA2 and have implications for the evaluation of breast cancer variants.

[1] Institut Curie, PSL Research University, UMR 3348, Genotoxic Stress and Cancer Unit, Research Center, Paris Sud University, Paris Saclay University, Centre Universitaire d'Orsay, Bâtiment 110, 91405 Orsay, France. [2] Department of Medicine, Imperial College London, SAF Building, London SW7 2AZ, UK. Correspondence and requests for materials should be addressed to A.C. (email: aura.carreira@curie.fr).

BRCA2-deficient tumour cells accumulate chromosomal abnormalities and this genome instability is thought to have a causative role in BRCA2-associated cancer[1,2]. BRCA2 maintains genome integrity through its function in the repair of DNA double-strand breaks by homologous recombination (HR)[3–5]. In this process, the recombination protein RAD51 and BRCA2, together with other factors, perform a reaction in which a damaged chromatid pairs with the intact sister resulting in the exchange of genetic information from the latter to the former. The DNA binding activity of BRCA2 is integral to its HR function as it facilitates the loading of the recombination protein RAD51 at DNA breaks. This function is ensured by the DNA binding domain located at the carboxy (C) terminus of the protein (CTD)[6]. Yet, cells resistant to DNA damage devoid of the entire CTD can still function in HR[7,8] suggesting that additional functional domains in BRCA2 could take over CTD's function. To test this hypothesis, we used protein secondary structure prediction tools (see Supplementary Methods) and identified a zinc finger (zf)-PARP like domain containing residues predicted to bind DNA in the amino (N) terminus of BRCA2 that are conserved in mammals (Supplementary Fig. 1). This analysis prompted us to test the functional relevance of this domain.

We reveal a DNA binding domain in the N terminus of BRCA2 that can stimulate RAD51-mediated homologous recombination and is mutated in breast cancer patients.

## Results

**The N terminus of BRCA2 binds DNA.** We expressed and purified from human cells several fragments of the N terminus of BRCA2; BRCA2$_{T1}$ (aa 1–250), BRCA2$_{T2}$ (aa 250–500), BRCA2$_{LT2}$ (aa 1–500) and BRCA2$_{LT3}$ (aa 1–750; Supplementary Fig. 1c,d) and tested their DNA binding activity by Electrophoretic Mobility Shift Assay (EMSA). Incubation of a ssDNA homopolymer, $dT_{40}$ and increasing concentrations of BRCA2$_{T2}$, BRCA2$_{LT2}$ or BRCA2$_{LT3}$, but not BRCA2$_{T1}$, generated a slower mobility species corresponding to DNA–protein complexes. Their DNA binding activity reached 20–40% of ssDNA–protein complex at 300 nM

(Fig. 1a,b). These results suggest that the region comprising BRCA2$_{T2}$ binds to ssDNA.

To further validate the DNA binding activity of this region, we examined the partition of BRCA2$_{T2}$ compared with BRCA2$_{T1}$ between biotinylated ssDNA ($dT_{80}$) immobilized on streptavidin magnetic beads challenged with excess $dT_{40}$ ssDNA. This experiment confirmed our results; only BRCA2$_{T2}$ and not BRCA2$_{T1}$ was titrated out by adding excess $dT_{40}$ indicating that BRCA2$_{T2}$ binds specifically to DNA (Supplementary Fig. 2a,b).

Collectively, these results indicate that the N-terminal region of BRCA2 binds DNA and that the region of aa 250–500 of BRCA2 (BRCA2$_{T2}$) comprising the putative zf-PARP domain is sufficient for this activity.

**BRCA2$_{T2}$ binds with stronger affinity to DNA than the CTD.** To examine the possible function and specificity of the DNA binding domain identified in the N-terminal region (NTD), we compared the DNA binding affinity of BRCA2$_{T2}$ with that of the CTD (aa 2,474–3,190; Fig. 2a) for different DNA substrates.

As described for the mouse CTD[6], human CTD bound to all DNA forms except dsDNA (Fig. 2b–f). Remarkably, at the attainable concentration (1.5 μM), BRCA2$_{T2}$ showed higher yield of DNA–protein complexes than the CTD for all DNA substrates tested (Fig. 2b–f). In contrast to the CTD, BRCA2$_{T2}$ also bound to dsDNA resulting in a yield of protein–dsDNA complex of ∼40% at 1.5 μM, similar to that of ssDNA (Fig. 2c, Supplementary Fig. 3a,b). Moreover, BRCA2$_{T2}$ exhibited significantly higher yield of DNA–protein complexes when bound to gapped DNA compared with the CTD, reaching a ∼40-fold difference at 1 μM of protein concentration (Fig. 2f).

In addition, we compared the DNA binding activity of BRCA2$_{T2}$ with that of full-length BRCA2. As expected, BRCA2 bound readily to ssDNA (Supplementary Fig. 4b) and to a lesser extent to dsDNA (Supplementary Fig. 4c) to levels comparable to those reported previously[3]. BRCA2$_{T2}$ bound ssDNA or dsDNA to the same extent as BRCA2 but the concentration to reach the same amount of product was ∼100 times higher suggesting that the two DNA binding domains, CTD and NTD, display

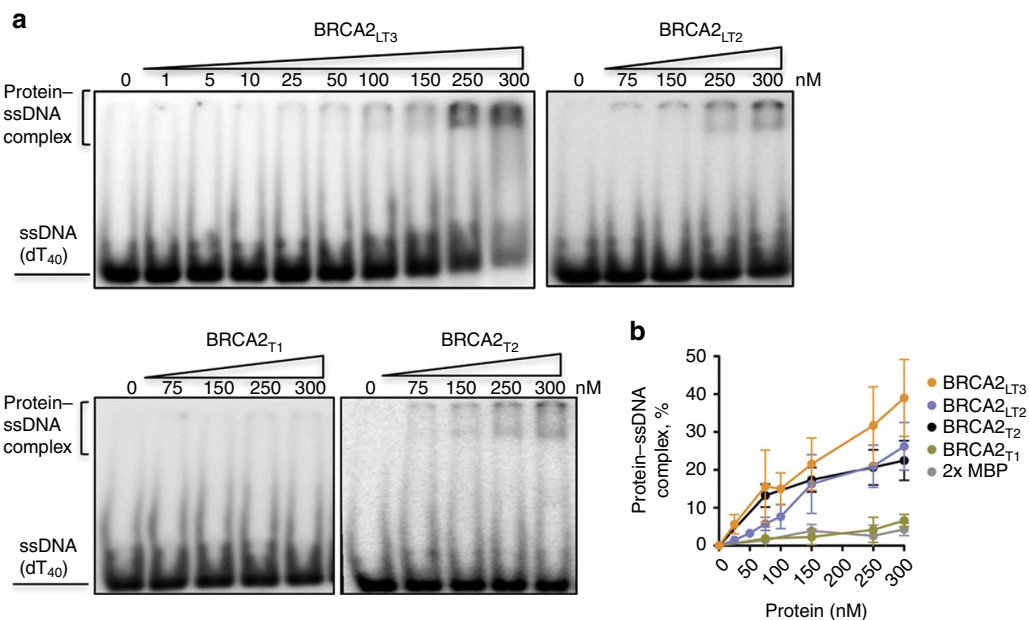

**Figure 1 | The N terminus of BRCA2 binds DNA. (a)** EMSA showing BRCA2$_{LT3}$, BRCA2$_{LT2}$, BRCA2$_{T1}$ and BRCA2$_{T2}$ binding to ssDNA ($dT_{40}$). **(b)** Quantification of **a**. Error bars, s.d. ($n = 3$).

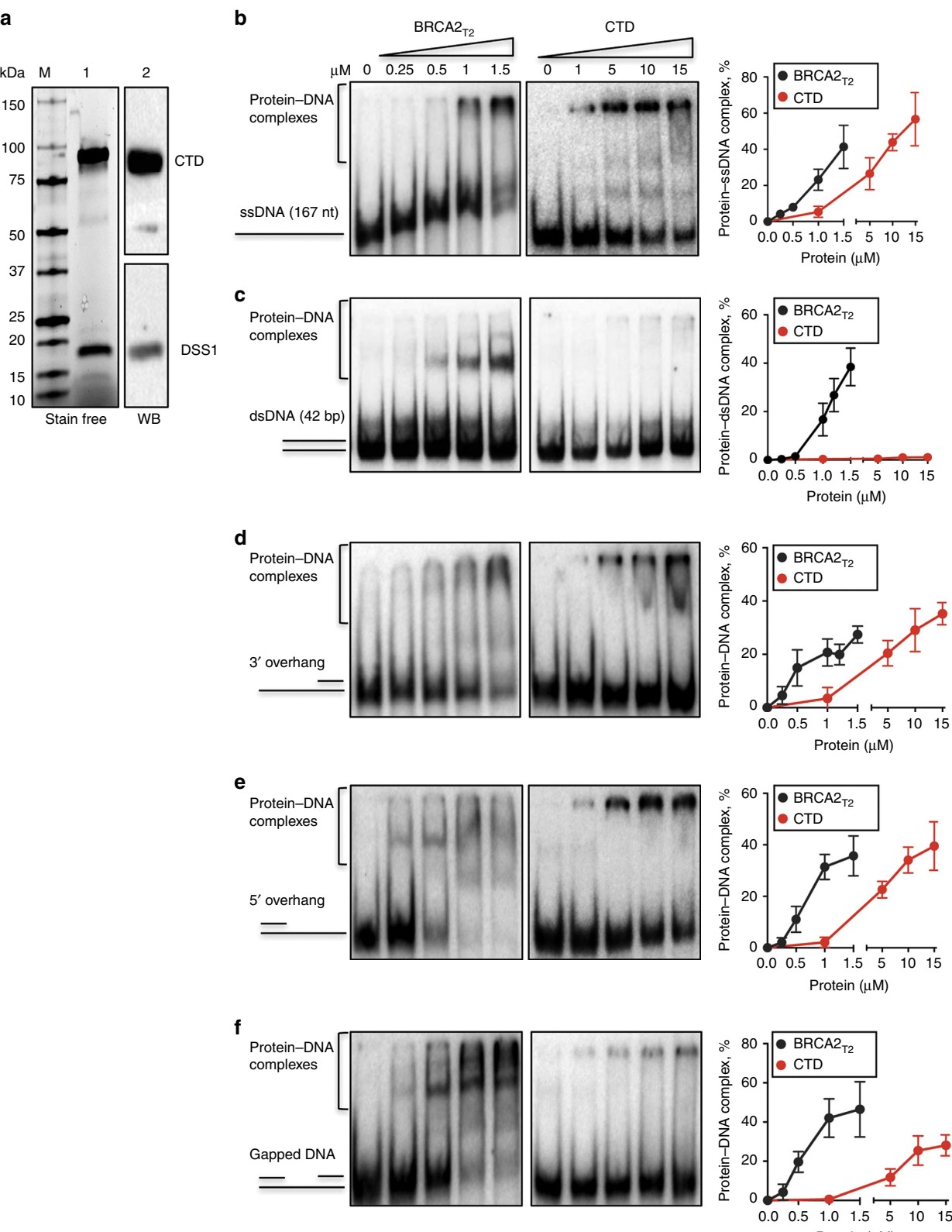

**Figure 2 | BRCA2_T2 binds to different DNA substrates and with stronger affinity than CTD.** (**a**) M size markers. (1) SDS–PAGE showing purified CTD (5 µg) loaded in a 4–15% acrylamide gel. (2) Western blot of the gel shown with an antibody against the His tag of the CTD and an antibody specific to DSS1 protein co-purified with it. (**b**) EMSA and quantification comparing the binding of BRCA2_T2 and CTD to ssDNA (167mer). (**c**) dsDNA (40mer), (**d**) 3′ ssDNA overhang, (**e**) 5′ ssDNA overhang, (**f**) gapped DNA. Error bars, s.d. (*n* = 3).

cooperative affinity for ssDNA in the context of the full-length protein. It is also possible that the isolated fragments display a different conformation than when comprised in the full-length protein explaining their lower affinity for DNA.

These results indicate that $BRCA2_{T2}$ exhibits an increased relative affinity or higher complex stability for all DNA substrates compared with the CTD, especially those containing dsDNA or a dsDNA/ssDNA junction.

**C315S breast cancer variant impairs NTD dsDNA binding.** NTD comprises a putative zf-fold, which is usually stabilized by cysteine and histidine residues[9,10]. To find out the residues important for DNA binding, we mutated three of the cysteines present in the domain (Supplementary Fig. 1) and tested their effect on DNA binding. We chose to mutate two cysteines that have been found substituted for serine in breast cancer patients (C315S, C341S) but for which the clinical relevance is still unknown (Breast Cancer Information Core and BRCAshare databases). The third one is a substitution of C279 to alanine (Fig. 3a). Single or double mutations were introduced in the $BRCA2_{T2}$ fragment and purified (Supplementary Fig. 5a). None of the single or double mutations affected significantly the ssDNA binding activity of $BRCA2_{T2}$ (Fig. 3b,c and Supplementary Fig. 5b,c). We next tested the effect of the single or the double mutations on dsDNA binding. Interestingly, the single substitution C315S reduced the dsDNA binding by ~6.5 fold, whereas the other single mutations did not or very mildly affected it (Fig. 3d,e). Accordingly, the double mutations containing the C315S substitution potentiated this effect (Fig. 3f,g). Moreover, C315S mutation mildly reduced the ability of $BRCA2_{T2}$ to bind a 3'overhang ssDNA by ~1.7 fold (Fig. 3h,i).

These results imply that C315S, a missense mutation frequently observed in breast cancer, highly impairs the interaction of BRCA2 with dsDNA.

**NTD can stimulate RAD51-mediated DNA strand exchange.** A fusion peptide containing a single BRC repeat of BRCA2 and the CTD or a BRC repeat and the ssDNA binding domain of replication protein-A (RPA), are able to enhance RAD51 recombination activity[11,12]. In the recombination process, replication RPA coats first the ssDNA to remove secondary structure, an interaction that is inhibitory for RAD51 assembly onto ssDNA due to the stronger affinity of RPA for ssDNA[13,14]. Mediator proteins like BRCA2 counteract the inhibitory effect of RPA. In view of the DNA binding activity exhibited by $BRCA2_{T2}$, we assessed whether a fusion of $BRCA2_{T2}$ and BRC4, $BRCA2_{BRC4-T2}$ (Supplementary Fig. 6a), could promote the DNA strand exchange activity of RAD51.

Optimal RAD51 (220 nM) and RPA (25 nM) concentrations based on RAD51 and RPA titration experiments (Supplementary Fig. 6b) were used in an *in vitro* short oligonucleotide DNA strand exchange assay[3]. RPA was incubated first with a 3'overhang ssDNA substrate (scheme in Fig. 4a), and as expected, RAD51 recombination activity was compromised (Fig. 4a, third lane). Importantly, $BRCA2_{BRC4-T2}$ was able to overcome the inhibition by RPA and stimulate the reaction by ~6-fold at 300 nM protein concentration reaching similar levels of stimulation of BRCA2 although at 10 times higher concentration (Fig. 4b). Strikingly, $BRCA2_{T2}$ alone also stimulated the reaction to a similar extent than $BRCA2_{BRC4-T2}$ (Fig. 4d and Supplementary Fig. 6c) though it does not interact directly with RAD51 (Supplementary Fig. 6d) or RPA (Supplementary Fig. 6e).

These results suggest that at saturating concentration of RAD51 and with a short DNA sequence, the binding of $BRCA2_{T2}$ to DNA is sufficient to facilitate RAD51 recombination. Another

explanation for this result could be that $BRCA2_{T2}$ counteracts the competition of RPA for ssDNA allowing assembly of RAD51 onto the ssDNA (Fig. 4c,d). Consistent with this idea, $BRCA2_{T2}$ did not stimulate the DNA strand exchange reaction in the absence of RPA (Supplementary Fig. 7a,b). In contrast, the fusion peptide $BRCA2_{BRC4-T2}$ showed a stimulatory effect of ~1.8 fold, comparable to that of full-length BRCA2 (Supplementary Fig. 7c,d). Similar to mouse CTD[6], human CTD did not stimulate the DNA strand exchange reaction in the absence of RPA (Supplementary Fig. 7a,b) but stimulated the RPA-containing reaction by ~3-fold at 10 μM (Fig. 4c,d).

In conclusion, $BRCA2_{BRC4-T2}$, $BRCA2_{T2}$ and, to a much lesser extent, the CTD can all promote the RPA-dependent DNA strand exchange activity of RAD51. It is important to note that the concentration of $BRCA2_{BRC4-T2}$ needed to reach the same amount of product formation as full-length BRCA2 is 10 times higher. These results can be explained by the fact that (i) the NTD alone binds with lower affinity to DNA than full-length BRCA2 (Supplementary Fig. 4b,c); (ii) the NTD is missing the DNA binding activity and stimulatory action of the CTD which is present in the full-length protein.

**dsDNA binding is required to stimulate HR at dsDNA/ssDNA.** Considering the binding of $BRCA2_{T2}$ to duplex DNA-containing structures, we reasoned that this activity might be required for the stimulation of DNA strand exchange when using dsDNA-containing substrates. We conducted a DNA strand exchange reaction using a 3' overhang ssDNA in the presence of the single mutant $BRCA2_{T2}$ C315S with reduced dsDNA binding activity. Indeed, $BRCA2_{T2}$ C315S decreased the stimulation of DNA strand exchange activity of RAD51 by ~3-fold compared with $BRCA2_{T2}$, an effect that was further potentiated by the double mutation C279A/C315S (Fig. 4e,f). To confirm that the defective stimulation of DNA strand exchange is due to the inability of $BRCA2_{T2}$ C315S to bind dsDNA and not to a faulty folding due to the mutation, we conducted the same reaction using a ssDNA substrate. Indeed, C315S mutation stimulated the ssDNA-based strand exchange of RAD51 as much as $BRCA2_{T2}$ (Fig. 4g,h) or full-length BRCA2 (Supplementary Fig. 8), suggesting that the single substitution C315S affects the stimulation of DNA strand exchange with the 3' ssDNA overhang substrate because of its defective dsDNA binding activity.

On the basis of the results with both DNA substrates, we can presume that $BRCA2_{T2}$ stimulate RAD51-mediated DNA strand exchange in two ways: (1) Physically blocking the binding of RAD51 to dsDNA and (2) Competing with RPA for ssDNA. Only the second would be acting in the case of C315S, which would explain why there is normal stimulation of DNA strand exchange when using a ssDNA substrate, whereas it is reduced in the case of the tailed DNA.

**Discussion**
A bias towards ssDNA/dsDNA junctions has been described for the functional homologues of BRCA2, the bacterial RecFOR, and its orthologue in *U. maydis*, Brh2, but not for human BRCA2 (ref. 3). Although the helix-turn-helix motif present in the CTD suggested dsDNA-binding activity[6], the interaction with this type of substrate has not been detected for CTD[6,15]. Our findings suggest that NTD specifically contributes the binding of BRCA2 to dsDNA allowing the association with dsDNA/ssDNA junctions.

Combining our results with previous findings[3–5,16–18] we propose a model whereby in the context of a ssDNA/dsDNA junction-containing lesion (Supplementary Fig. 9); NTD binds at the dsDNA/ssDNA junction first, facilitating the loading and

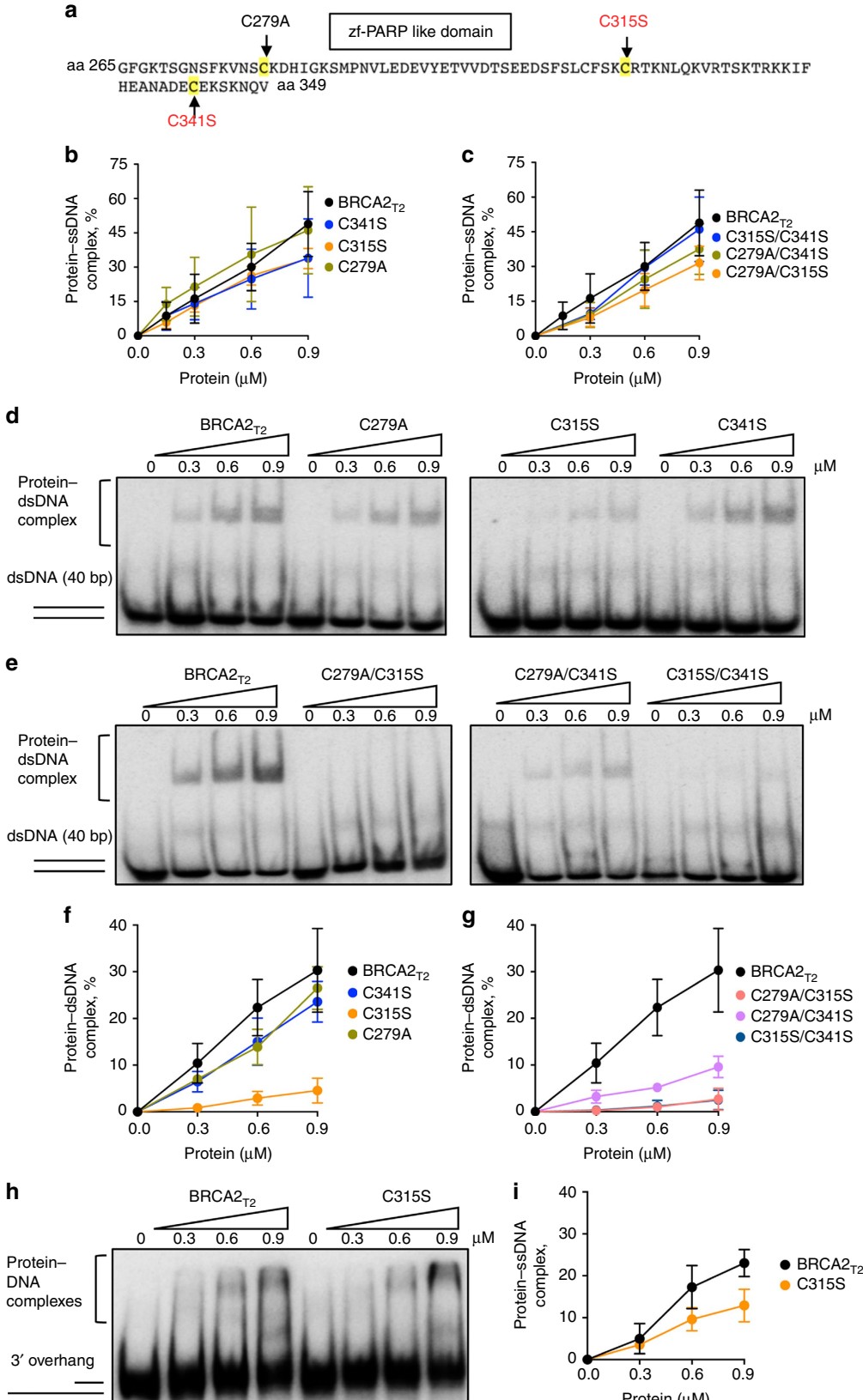

**Figure 3 | Cysteine residues located in the putative zf-PARP-like domain and mutated in breast cancer patients affect the dsDNA binding activity of BRCA2$_{T2}$.** (**a**) amino-acid sequence comprised in the putative zf-PARP-like domain defined by SMART showing the mutated cysteine residues. The ones found in breast cancer patients are highlighted in red. (**b**) Quantification of EMSA displayed in Supplementary Fig. 5b, (**c**) showing the binding of BRCA2$_{T2}$, the indicated single mutants (**c**), and the double mutants to ssDNA. (**d**) EMSA showing the binding of BRCA2$_{T2}$ and the indicated single mutants (**e**), and the double mutants to dsDNA. (**f**) Quantification of **d**. (**g**) Quantification of **e**. (**h**) EMSA showing the binding of BRCA2$_{T2}$ and BRCA2 C315S to 3' overhang ssDNA. (**i**) Quantification of **h**. Error bars, s.d. ($n = 3$).

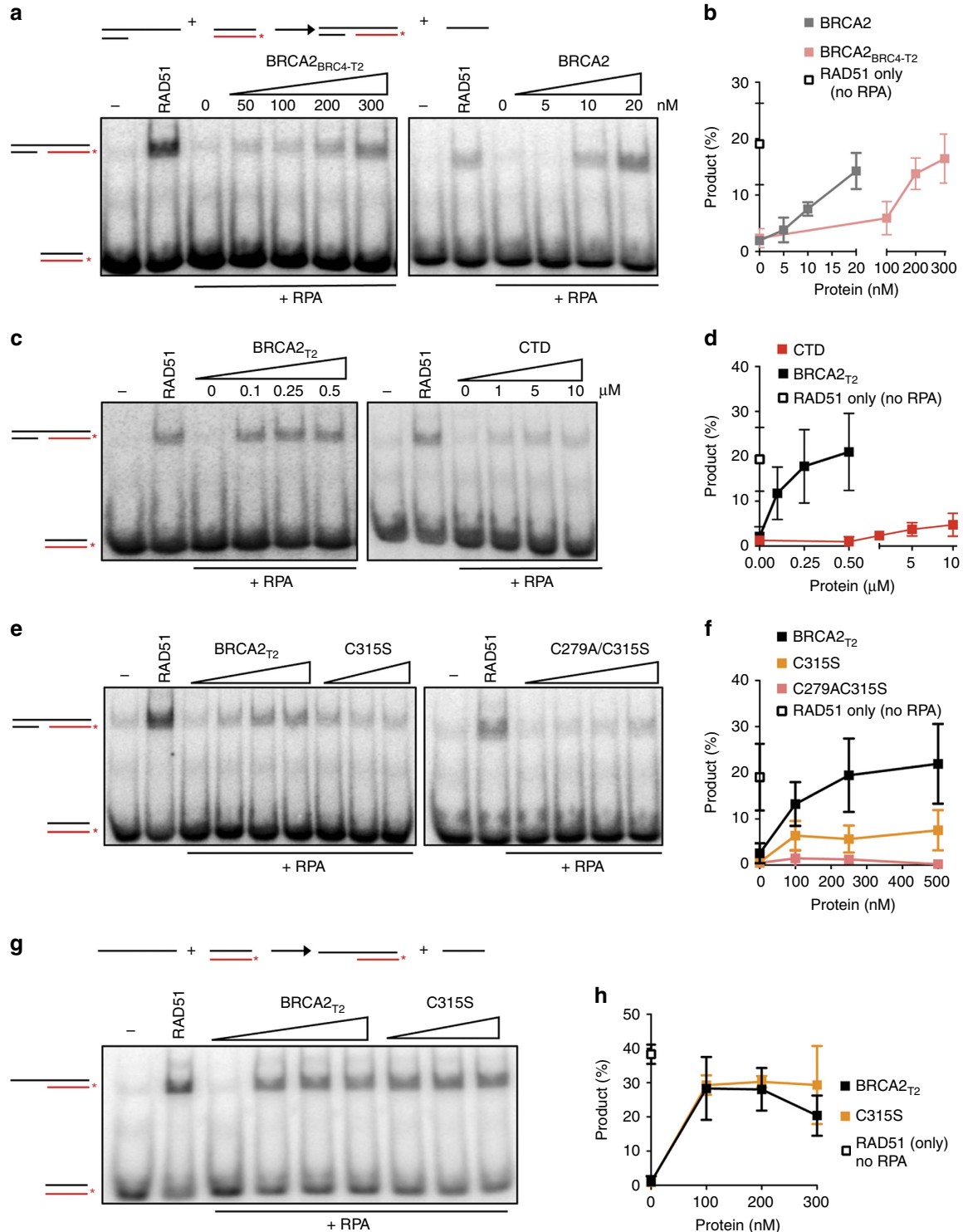

**Figure 4 | BRCA2$_{T2-BRC4}$ and BRCA2$_{T2}$ stimulate RAD51-promoted DNA strand exchange.** (**a**) DNA strand exchange reaction using a 3' overhang ssDNA substrate (scheme on top) in the presence or absence of RPA, RAD51, and increasing concentrations of BRCA2$_{BRC4-T2}$ or BRCA2. (**b**) Quantification of (**a,c**), same reaction as in **a** with increasing concentrations of BRCA2$_{T2}$ or CTD. (**d**) Quantification of (**c,e**), same reaction as in **a** with increasing concentrations of BRCA2$_{T2}$ or BRCA2$_{T2}$ mutant in C315S, or C279A/C315S, as indicated. (**f**) Quantification of (**e,g**), same reaction as in **a** using a ssDNA substrate (scheme on top) with increasing concentrations of BRCA2$_{BRC4-T2}$ or BRCA2$_{T2}$ mutant C315S, as indicated. (**h**) Quantification of **g**. Error bars, s.d. ($n = 3$).

stabilization of RAD51 nucleoprotein filament onto RPA-coated ssDNA by the BRC repeats[18]. CTD binds along the ssDNA and actively facilitates the displacement of RPA[19] allowing multiple nucleation events and filament extension of

RAD51. These activities enhance the subsequent steps of homologous recombination.

It is possible that both NTD and CTD are required for assembly of RAD51 onto ssDNA in the context of full-length

BRCA2. Alternatively, the two DNA binding sites may act on different substrates depending on the DNA damage encountered. How the two DNA binding modules of BRCA2 are coordinated *in vivo* warrants further investigation.

## Methods

**Expression and purification of proteins.** All BRCA2 N-terminal expression constructs containing the sequence coding for BRCA2 amino acids 1–250 (BRCA2$_{T1}$), 251–500 (BRCA2$_{T2}$), 1–500 (BRCA2$_{LT2}$) and 1–750 (BRCA2$_{LT3}$) were amplified by PCR from a 2 × MBP-BRCA2 phCMV1 vector[3] and cloned by restriction digest with XhoI and NotI into a phCMV1 vector containing an N-terminal 2 × MBP tag with two nuclear localization signals (NLS) downstream the tag.

Point mutations were introduced using QuikChange II site-directed mutagenesis kit (Agilent) and verified by sequencing. BRCA2$_{BRC4-T2}$ fusion construct was generated by HIFI Gibson Assembly (NEB) in the phCMV1 2 × MBP 2 NLS vector.

All the constructs mentioned above including the vector containing the MBP tag and the NLS used as control were amplified with NucleoBond Xtra Midi kit (Macherey Nagel) and used for transfection.

Ten 150 mm confluent plates of HEK293 cells were transiently transfected with TurboFect (Thermo Scientific) following the manufacturer's specifications and harvested 30 h post transfection. After lysis with 50 mM HEPES (pH 7.5), 250 mM NaCl, 5 mM EDTA, 1% NP40, 1 mM DTT, 1 mM PMSF and EDTA-free Protease Inhibitor Cocktail (Roche), the suspension was incubated for 3 h with amylose resin (NEB). The 2 × MBP tagged BRCA2 fragments were eluted with 10 mM maltose. The eluates were further purified with Bio-Rex 70 cation-exchange resin (Bio-Rad) by NaCl step elution (50 mM HEPES pH 7.5, 0.5 mM EDTA, 10% glycerol, 1 mM PMSF, 1 mM DTT and 250 mM, 450 mM or 1 M NaCl).

The size and purity of the BRCA2 fragments were verified by loading the final fractions on a SDS–PAGE (polyacrylamide gel electrophoresis) gel and detecting the MBP tag by western Blot (mAB R29, Invitrogen).

The protein concentration of the nuclease-free fractions was determined using NanoOrange Protein Quantitation Kit (Thermo Fisher) and by density determination in SDS–PAGE gel using BSA as a standard.

The C-terminal DNA binding domain (aa 2,474–3,190) of human BRCA2 (CTD) was cloned into pET28 6His SUMO vector. pCDF DSS1 expression vector was generated by HIFI Gibson Assembly (NEB) from a pFASTBAC Dual-DSS1 vector (kind gift from R.B. Jensen). The CTD was co-expressed with DSS1 to ensure stability of the protein. *E.coli* BL21 DE3 pISO Dscb cells (kind gift of A. el Marjou) were transformed and grown at 37 °C in 7 litres of Terrific Broth and induced with 0.5% arabinose and 1 mM IPTG overnight at optical density 2. The cells were collected in 20 mM Tris-HCl pH 8, 300 mM NaCl, 10% glycerol, 0.5 mM EDTA, 5 mM β-mercaptoethanol, 1 × Protease Inhibitor Cocktail EDTA-free (Roche), 10 mM MgCl$_2$, 1 × DNase, 0.5 mg ml$^{-1}$ Lysozyme (Sigma-Aldrich) and the suspension was lysed by disintegration at 1.7 kbar.

The His-tagged protein was incubated with Protino Ni-NTA agarose (Macherey Nagel) and eluted with 200 mM imidazole. After dialysis overnight against 20 mM Tris-HCl pH 8, 100 mM NaCl, 10% glycerol, 5 mM β-mercaptoethanol, the eluate was loaded onto a 5 ml HiTrap Heparin HP column (GE) and eluted using a continuous NaCl gradient (100-1 M NaCl) in the same buffer.

The protein was visualized on a 7.5% SDS gel and the protein concentration was determined by Bradford assay.

To generate EGFP-MBP-BRCA2, one MBP-tag of a phCMV1 2 × MBP-BRCA2 vector (kind gift from S.C. Kowalczykowski) was substituted by an EGFP tag amplified by PCR from a pEGFP-C1 vector (Invitrogen) and cloned by restriction digest with KpnI. EGFP-MBP-tagged BRCA2 was purified as described above for the protein fragments. The concentration was determined by NanoOrange Protein Quantitation Kit (ThermoFisher Scientific) and adjusted by loading it on a gel, subtracting the contributions from contaminants or truncated BRCA2 polypeptide.

Human RAD51 was cloned into pCDF 6his SUMO vector. *E. coli* BL21 BRL were transformed and grown in 7 l of Terrific Broth at 37 °C in a fermenter. At optical density = 1, the temperature was decreased to 20 °C and protein expression was induced with 0.5 mM IPTG overnight at 700 RPM. The cells were harvested by centrifugation at 9,000g for 10 min at 4 °C and resuspended in 1 × PBS, 350 mM NaCl, 20 mM imidazole, 10% glycerol, 0.5 mg ml$^{-1}$ lysozyme, 1 × Protease Inhibitor Cocktail (Roche), 0.5 mM PMSF, 10 mM MgCl$_2$, DNase and extracted using a disintegrator at 1.8 kbar and collected by centrifugation at 48,000g for 30 min. The His-tagged protein was incubated with Protino Ni-NTA agarose (Macherey Nagel) and washed with 20 mM Tris-HCl pH 8.0, 20 mM imidazole, 10% glycerol, 100 mM NaCl, 5 mM β-mercaptoethanol. The His-tag was cleaved by incubating the resin with 0.4 mg ml$^{-1}$ SUMO Protease at 4 °C overnight. The supernatant from the Ni-NTA resin containing the cleaved RAD51 protein was then loaded onto a 5 ml HiTrap Heparin HP chromatography column (GE) in 20 mM Tris-HCl pH 8, 10% glycerol, 100 mM NaCl and 5 mM β-mercaptoethanol. RAD51 protein was eluted using a continuous NaCl gradient (100 mM-2M NaCl) in the same buffer. The eluate was dialysed against RAD51 storage buffer (20 mM Tris HCl pH 8, 50 mM KCl, 0.5 mM EDTA, 1 mM DTT, 10% glycerol, 0.5 mM PMSF) and visualized on a 7.5% SDS–PAGE gel stained with Coomassie Brilliant Blue. The protein concentration was determined by Bradford assay.

His-tagged RPA was a kind gift from M. Modesti (IGH, Marseille).

BRCA2$_{T2-BRC4}$ fusion construct was generated by Gibson HIFI assembly (NEB) in the phCMV1-EGFP-MBP 2NLS BRC4 vector background (pAC138).

After purification, all proteins and fragments were aliquoted, frozen in liquid nitrogen and stored at − 80 °C.

**Antibodies.** Antibodies were obtained from the following sources: Anti-BRCA2 (1:2,000; OP95-100UG, EMD Milipore), anti-DSS1 (1:500; sc-28848, Santa Cruz Biotechnology), anti-MBP (1:5,000; 33–5,100, Invitrogen), anti-His (1:1,000; PEP-156P Eurogentec).

Uncropped western blots for Fig. 2, and Supplementary Figs 1 and 4 are shown in Supplementary Fig. 10.

**Electrophoretic mobility shift assay.** DNA substrates were purchased PAGE-purified from MWG Eurofins. The sequence of the oligonucleotides used for these assays are included in Supplementary Table 1. The ssDNA substrates used in EMSA were oAC379 and oAC423 $^{32}$P labelled at the 5'-end. To generate the dsDNA substrates, oAC405 was $^{32}$P labelled at the 5'-end and annealed in a 1:1 molar ratio to oAC406. The 3' and 5' overhang substrates were produced by annealing $^{32}$P-labelled oAC490 (42mer − 3') to oAC423 (167mer) or oAC403 (42mer − 5') to oAC423 in a 1:1 molar ratio, respectively. oAC423, oAC403 and oAC490 were annealed in a 1:1:1 ratio to produce the gapped DNA substrate.

The proteins were incubated at the indicated concentrations with 0.2 μM nucleotides $^{32}$P-labelled DNA substrates for 1 h at 37 °C in EMSA reaction buffer (25 mM Tris Acetate pH 7.5, 1 mM DTT, 1 mM MgCl$_2$, 2 mM CaCl$_2$). The protein-DNA complexes were resolved on 6% native polyacrylamide gels in 1 × TAE buffer (40 mM Tris Acetate, 0.5 mM EDTA) at 70 V for 75 min. The gels were dried and analysed with a Typhoon PhosphorImager (Amersham Biosciences) using Image Quant software (GE Healthcare). In all EMSAs except for the ones shown in Supplementary Fig. 4, the ratio of DNA–protein complexes was calculated as the percentage of bound DNA compared with the free DNA. Because full-length BRCA2 binding to DNA results in fast and slow complexes that result in a smear, the percentage of protein–DNA complexes in Supplementary Fig. 4 was quantified as the free radiolabelled DNA remaining in a given lane relative to the protein-free lane. The protein-free lane defined the value of 0% complex.

**DNA strand exchange assay.** The 3'tail DNA substrate and dsDNA donor template were generated as described for the EMSA except that only the dsDNA was radiolabelled. RPA (25 nM) or storage buffer was pre-incubated with 4 nM molecules of 3'tail DNA or ssDNA for 5 min at 37 °C. Then, RAD51 (220 nM) alone or with the indicated concentrations of BRCA2 were added to the mix and incubated for 5 min at 37 °C in a buffer containing 25 mM Tris Acetate pH 8.0, 1 mM DTT, 2 mM ATP, 1 mM MgCl$_2$, 2 mM CaCl$_2$, 0.1 mg ml$^{-1}$ BSA. The reaction was started by adding 4 nM molecules of the donor template dsDNA (oAC405 and oAC406 1:1) and the mix was further incubated for 30 min at 37 °C. The reaction was stopped by incubation with 0.25% SDS and 0.5 mg ml$^{-1}$ Proteinase K for 10 min. The samples were loaded on a 6% polyacrylamide gel and migrated at 70 V for 75 min. The gels were dried and analysed with a Typhoon PhosphorImager (Amersham Biosciences) using Image Quant software (GE Healthcare). The percentage of DNA strand exchange product was calculated as labelled product divided by total labelled input DNA in each lane.

**Affinity pull-down with amylose beads.** Before pull-down assays, amylose resin (NEB) was equilibrated with binding buffer: 50 mM HEPES (pH 7.5), 250 mM NaCl, 0.5 mM EDTA and 1 mM DTT. Purified 2XMBP-BRCA2$_{BRC4-T2}$ or 2xMBP-BRCA2$_{T2}$ proteins (1.25 μg) were incubated with 1 μg purified RAD51 for 30 min at 37 °C and then batch bound to 30 μl of amylose resin for 1 h at 4 °C. The complexes were washed with binding buffer (50 mM HEPES (pH 7.5), 250 mM NaCl, 0.5 mM EDTA and 1 mM DTT) containing 1% NP-40. Proteins were eluted with 30 μl 10 mM maltose in binding buffer, resuspended in 1 × SDS sample buffer, heated at 54 °C for 5 min and loaded onto a 4–15% gradient SDS–PAGE gel. The gels were stained with SYPRO Orange.

**Data availability.** All the relevant data are available from the authors upon request.

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

## Acknowledgements

This work was supported by the ATIP-AVENIR CNRS/INSERM Young Investigator grant 201201, EC-Marie Curie Career Integration grant CIG293444, FRM 'Amorçage Jeunes Equipes' Young Investigator grant AJE201101, Institut National du Cancer INCa-DGOS_8706 to A.C. and Wellcome Trust 098412/Z/12/Z to X.Z. C.N. was supported by a PhD Fellowship from Institut Curie and FRM Fellowship FDT20150532491 for the fourth year of PhD. We would like to thank Sophie Zinn-Justin for helpful discussions and editorial input. We are grateful to Ahmed El Marjou and Patricia Duchambon for their help in CTD and RAD51 expression and purification.

## Author contributions

A.C. conceived the general ideas for this study. C.v.N. and A.C. planned the experiments and interpreted the data; C.v.N., A.E. and C.M. performed the experiments. A.C. wrote the manuscript with contributions from X.Z. and C.v.N.

## Additional information

**Competing financial interests:** The authors declare no competing financial interests.

