## [Peer review file · Nature Communications]

Reviewers' Comments:

Reviewer #1 (Remarks to the Author)

Aura Carreira, Catharina von Nicolai, Åsa Ehlén, Taha Shahid, Yueru Sun, Charlotte Martin, and Xiaodong Zhang

The authors report the identification of a second DNA binding domain in the N-terminal region of BRCA2 besides the previously noted C-terminal DNA binding domain. While an N-terminal DNA binding domain was identified in the *Ustilago* BRCA2 homolog Brh2, this finding is nevertheless somewhat unexpected, as it had been widely speculated that the N-terminal BRCA2 interaction factor PLAB2 supplanted the N-terminal DNA binding site. The identification of this DNA binding domain is an important contribution that advances our understanding of this important tumor suppressor protein.

Figure 1: EMSA experiments establish that the N-terminal DNA binding domain is located between amino acids 250 and 500. However, the quality of the EM imaging appears to be poor. Gapped NDA should be shown with the same magnification. The particles of BRCA2 LT3 shown in c) appear to be smaller than those in Extended Figure 2a. Is the calculated volume in c) consistent with the size of BRCA2 LT3? The imaging with the antibody (part e) is quite unclear, and the images, as presented, do not allow a conclusion. The complete absence of quantitation is noted.

In extended Fig. 2a, the particles of BRCA2LT3 show a wide distribution in particle size, from smaller particles (degradation?) to larger aggregates as labeled. It is uncertain how the authors determine the boxed particles are BRCA2LT3, and what evidence is there to support. Considering extended Fig. 2c, it seems that only the smaller degradations (smaller particles) bind to DNA, which is consistent with a decrease in volume.

Figure 2: EMSA experiments compare the DNA binding properties of the isolated N-terminal (1-500) and C-terminal DNA binding domains. The data show that the N-terminal domain can bind dsDNA, whereas the C-terminal cannot, consistent with the data reported by Yang et al. for the C-terminal domain. The results with LT2 show quite a difference to the data reported in Fig. 1a (25% binding at 300 nM versus under 10% binding at 500 nM in Figure 2b). This and other inconsistencies (see below) remain unexplained.

These DNA binding properties should be directly compared with the full-length protein, which appears to be available to the authors (Extended Data Figures 5, 6). The quality of the full-length protein should be documented. It appears that the isolated domains show a significant lower affinity than the full-length protein.

Figure 3: The authors compare the DNA binding affinities of wild type mutant BRCA2 N-terminal fragments. It is unclear, whether it was shown that these mutations are disease causing or disease associated.

Compared to Figures 1 and 2, there are inconsistencies in the DNA binding data that should be addressed. The binding in 3b/c appears much higher than in Figures 1 and 2. Also in extended Fig. 4b & c, the binding of ssDNA by BRCA2T2 should be identical. However, in b, all ssDNA get bound by 0.9 μ M BRCA2T2, but in c, almost all ssDNA is still free and not bound by 0.9 μ M BRCA2T2.

The key result is the reported dsDNA binding defect of the C315A mutant protein. However, there

is a problem. Fig. 3H, C315S has much less free ssDNA left at 0.9 μ M, than BRCA2T2 at 0.9 μ M, indicating a better binding of C315S. However, the quantitation in Fig. 3I shows the opposite. Is there a nuclease/phosphatase contamination in C315S?

The x-axes in F and G are mislabeled.

Figure 4: Fig. 4a-d, the observation that BRCA2T2 stimulates Rad51-catalyzed strand exchange in the presence of RPA inhibition much better than the construct BRCA2BRC4-T2 lead the authors to conclude that "BRCA2T2 counteracts the competition of RPA for ssDNA allowing assembly of RAD51 onto the ssDNA". However, in Fig. 4e-h, the testing of BRCA2T2 C315S is to test whether dsDNA binding is required. Did the author want to claim that both ssDNA and dsDNA binding is required? The data is more consistent with the idea that BRCA2T2 is simply competing with RPA for DNA binding and thus increase Rad51-catalyzed strand exchange, since RAD51 interaction is not even needed.

Did the authors test binding to RPA? There is some discrepancy in reports about BRCA2-RPA interaction. Jensen et al. reported no interaction of the full-length protein, whereas Wong et al. described a specific interaction between human BRCA2 and RPA.

Additional points:

- 1) Page 2, line 50: It is not clear, why the authors propose BRCA2 specificity for loading RAD51 to junctions, when the full-length protein does not bind junction better than ssDNA and shows only about two-fold better activity with junction substrates in DNA strand exchange reactions.
- 2) Page 5, line 106: It is unclear, what the wording 'artificial' means in this context.
- 3) Page 6, line 128: Please give the optimal concentrations of RAD51 and RPA used in the experiments.

Reviewer #2 (Remarks to the Author)

The paper is describing the existence of a DNA binding domain in the N-terminus of BRCA2, which has not previously been identified. Further, the DNA binding characteristics of this domain show preference for dsDNA and ds/ss DNA interfaces. There is contrast with the DNA binding domain of the C-terminal domain of BRCA2, which shows more ssDNA and 3'-overhang preference. Putting this new information together, it suggests that BRCA2 acts as a mediator for RAD51 by binding both dsDNA and ssDNA either side of a transition to allow RAD51 nucleation.

The evidence presented looks to be unveiling an interesting new twist to understanding the function of BRCA2. It should be noted that the biochemistry of the strand exchange reaction requires 10-fold more BRCA2-NTD fragment (T2 = aa250-500). I think this difference in stoichiometry is probably not unexpected, but should be discussed in the description of the model.

The importance of the cysteines in this T2 fragment is emphasized by the effect of mutating one or two of the 3 cysteines in this fragment. In particular, the C315S mutant removes a significant amount of the activity.

Overall, there appears to be novel findings that will be of interest to the BRCA2 community.

Reviewers' comments:

Reviewer #1 (Remarks to the Author):

Aura Carreira, Catharina von Nicolai, Åsa Ehlén, Taha Shahid, Yueru Sun, Charlotte Martin, and Xiaodong Zhang

The authors report the identification of a second DNA binding domain in the N-terminal region of BRCA2 besides the previously noted C-terminal DNA binding domain. While an N-terminal DNA binding domain was identified in the Ustilago BRCA2 homolog Brh2, this finding is nevertheless somewhat unexpected, as it had been widely speculated that the N-terminal BRCA2 interaction factor PLAB2 supplanted the N-terminal DNA binding site. The identification of this DNA binding domain is an important contribution that advances our understanding of this important tumor suppressor protein.

Figure 1: EMSA experiments establish that the N-terminal DNA binding domain is located between amino acids 250 and 500. However, the quality of the EM imaging appears to be poor. Gapped NDA should be shown with the same magnification. The particles of BRCA2 LT3 shown in c) appear to be smaller than those in Extended Figure 2a. Is the calculated volume in c) consistent with the size of BRCA2 LT3? The imaging with the antibody (part e) is quite unclear, and the images, as presented, do not allow a conclusion. The complete absence of quantitation is noted.

In extended Fig. 2a, the particles of BRCA2LT3 show a wide distribution in particle size, from smaller particles (degradation?) to larger aggregates as labeled. It is uncertain how the authors determine the boxed particles are BRCA2LT3, and what evidence is there to support. Considering extended Fig. 2c, it seems that only the smaller degradations (smaller particles) bind to DNA, which is consistent with a decrease in volume.

We thank the reviewer for these detailed comments. We recognize the quality issues in these images and taking the suggestion of this reviewer we have decided to remove the EM data from the manuscript. We think however that because these are supporting results the conclusions of the manuscript are not affected by the removal of the images as the DNA binding activity is extensively demonstrated with the biochemical assays. In addition, we have now included supporting data where we show the DNA binding specificity of T2 fragment but not T1 using biotinylated DNA. This binding can be competed out by excess non biotinylated DNA (Extended Data Figure 2), further validating our results.

Figure 2: EMSA experiments compare the DNA binding properties of the isolated N-terminal (1-500) and C-terminal DNA binding domains. The data show that the N-terminal domain can bind dsDNA, whereas the C-terminal cannot, consistent

with the data reported by Yang et al. for the C-terminal domain. The results with LT2 show quite a difference to the data reported in Fig. 1a (25% binding at 300 nM versus under 10% binding at 500 nM in Figure 2b). This and other inconsistencies (see below) remain unexplained.

We thank the reviewer for this comment. There is indeed a difference in binding which is due to the different substrate utilized; this is now specified in each figure. In the case of Figure 1, the DNA binding was done with a homopolymer (dT40), whereas in Figure 2, the DNA binding was performed with a 167mer composed of mixed sequence. A similar preference for T (or C) was observed for the C-terminal DNA binding domain of BRCA2 (Yang et al., 2002).

full-length protein, which appears to be available to the authors (Extended Data Figures 5, 6). The quality of the full-length protein should be documented. It appears that the isolated domains show a significant lower affinity than the full-length protein.

To address this comment we have now included a new figure, Extended Data Figure 4. First, the purified BRCA2 protein is shown in a SDS-PAGE gel and the identity of the protein is validated by Western Blot (Extended Data Fig. 3a). Second, the DNA binding ability of T2 was directly compared to that of full length BRCA2 using the same DNA substrates as were used in the original paper where we biochemically characterized human BRCA2 (Jensen *et al.*, Nature 2010). As shown in Extended Figure 4b, the ssDNA and dsDNA binding affinity of the full-length protein are higher than the one of T2 fragment. Because there are two ssDNA binding domains in the full-length BRCA2, NTD and CTD, the higher affinity for ssDNA observed in the full-length protein might reflect a cooperativity between the two DNA binding domains. It is also possible that the conformation of the isolated fragments is slightly different than in the context of the full-length protein explaining their lower affinity for DNA.

Figure 3: The authors compare the DNA binding affinities of wild type mutant BRCA2 N-terminal fragments. It is unclear, whether it was shown that these mutations are disease causing or disease associated.

The mutations C315S and C341S are listed in the BRCShare and BIC databases. In particular, C315S has been reported in 19 breast cancer patients. However, although these mutations have been identified in patients, they are too rare to be classified based on genetic data so they are listed as "variants of unknown clinical significance (VUS)". We think functional assays such as the one described here (DNA binding activity) will help evaluate the causality of these variants (Guidugli *et al.*, 2014 Human Mutation). We have now modified the text so that this point is clarified in page 5.

Compared to Figures 1 and 2, there are inconsistencies in the DNA binding data that should be addressed. The binding in 3b/c appears much higher than in Figures 1 and 2.

In figure 1 the maximum concentration utilized is 300 nM of BRCA2_{T2}. The ssDNA binding activity at that concentration is ~20%. In Figure 3B and 3C, the DNA binding

activity at 300 nM of BRCA2_{T2} is ~15%. The slight differences between the two is most likely due to experimental variability.

Also in extended Fig. 4b & c, the binding of ssDNA by BRCA2T2 should be identical. However, in b, all ssDNA get bound by 0.9 μM BRCA2T2, but in c, almost all ssDNA is still free and not bound by 0.9 μM BRCA2T2.

We thank the reviewer for this remark. Indeed, we detected an issue with the ssDNA substrate used in some of these experiments and we decided to repeat all the sets including BRCA2_{T2}, the single and double mutants with a new DNA substrate. Here, the ssDNA binding activity of BRCA2_{T2} is consistent with that in Figure 1, where we used the same substrate. Importantly, the single and double mutants bind this substrate to the same extent as BRCA2_{T2}. This is not the case for the dsDNA for which there is a clear difference for the mutants versus BRCA2_{T2} wt (Figure 3 d-g).

We acknowledge this is an important point as it was not the same as reported in the first version of the manuscript, however, it does not modify our conclusions. We have now substituted the graphs in figure 3b, 3c, the EMSAs in Extended Figure 5b and c, and the text, accordingly.

The key result is the reported dsDNA binding defect of the C315A mutant protein. However, there is a problem. Fig. 3H, C315S has much less free ssDNA left at 0.9 μM, than BRCA2T2 at 0.9 μM, indicating a better binding of C315S. However, the quantitation in Fig. 3I shows the opposite. Is there a nuclease/phosphatase contamination in C315S?

The dsDNA binding activity of C315S is shown in Figure 3d and quantified in Figure 3f, which shows a clear reduction in both the graph and the gel. Figure 3h shows the DNA binding activity of C315S to 3'-tail DNA. We acknowledge the fact that the quantification might not match with the gel and we did a new set of experiments to validate the quantification. The new EMSAs and quantifications are shown now in the new figure 3h, 3i. As we have described in the first version of the manuscript, the new results show only a moderate reduction in the 3'-tail DNA binding.

The x-axes in F and G are mislabeled.

We thank the reviewer; this is now corrected in the new version of the manuscript.

Figure 4: Fig. 4a-d, the observation that BRCA2T2 stimulates Rad51-catalyzed strand exchange in the presence of RPA inhibition much better than the construct BRCA2BRC4-T2 lead the authors to conclude that "BRCA2T2 counteracts the competition of RPA for ssDNA allowing assembly of RAD51 onto the ssDNA".

In contrast to the reviewer comment, we do not observe a big difference in the stimulation of DNA strand exchange by BRCA2_{T2} versus BRCA2_{BRC4-T2}. This is now clarified in a new graph where we compare side by side the stimulation of DNA strand exchange by BRCA2_{T2} and BRCA2_{BRC4-T2}. As shown in the new graph presented in Extended Data Fig. 6c, the stimulation of both fragments is very similar, which is what we stated in the manuscript.

However, in Fig. 4e-h, the testing of BRCA2T2 C315S is to test whether dsDNA

binding is required. Did the author want to claim that both ssDNA and dsDNA binding is required? The data is more consistent with the idea that BRCA2T2 is simply competing with RPA for DNA binding and thus increase Rad51-catalyzed strand exchange, since RAD51 interaction is not even needed.

Figure 4e shows that C315S, the mutant that reduces the dsDNA binding activity of BRCA2T2, affects the stimulation of DNA strand exchange of RAD51 so that the level of product formation is reduced by almost 3-fold. We then hypothesized that if the dsDNA binding activity is what is making the difference in stimulation, we should not see a difference in product formation when we try the DNA strand exchange assay using a ssDNA substrate. This is indeed what we found as it is shown in Fig. 4g and quantified in Fig. 4h. Thus, at least in the context of the tailed DNA substrate, we conclude that the dsDNA binding activity might be required for an optimal stimulation of DNA strand exchange. As BRCA2_{T2} binds readily to dsDNA and the dsDNA/ssDNA junction, we presume that wt BRCA2_{T2} is acting in this substrate in two ways, 1. blocking the binding of RAD51 to dsDNA, 2. Competing with RPA for ssDNA and therefore facilitating RAD51-mediated strand exchange as the reviewer stated. In the case of the ssDNA substrate, the dsDNA binding of BRCA2_{T2} is not required so activity 1. is not present, which explains that both the mutant and the wt BRCA2_{T2} stimulate the reaction to the same extent. We have rephrased this paragraph in the manuscript so that is more clear to the reader.

Did the authors test binding to RPA? There is some discrepancy in reports about BRCA2-RPA interaction. Jensen et al. reported no interaction of the full-length protein, whereas Wong et al. described a specific interaction between human BRCA2 and RPA.

As the reviewer said, we have reported before that full-length BRCA2 does not bind to RPA (Jensen et al., 2010). The interaction with RPA was shown for another region of the N-terminus (around aa 34) of BRCA2 not included in BRCA2_{T2} (aa 250-500) however, as requested, we tested the interaction of BRCA2_{T2} to RPA using amylose pull-down. Although there was some non-specific binding to the beads as reflected in the gel, BRCA2_{T2} did not bind RPA, consistent with the data of the full-length protein. These results are now included as Extended Data Fig. 6e.

Additional points:

1) Page 2, line 50: It is not clear, why the authors propose BRCA2 specificity for loading RAD51 to junctions, when the full-length protein does not bind junction better than ssDNA and shows only about two-fold better activity with junction substrates in DNA strand exchange reactions.

Given the unique dsDNA binding activity of BRCA2_{T2}, and the higher affinity for gapped-DNA compared to the CTD (Figure S3), we think that in the context of a recombination intermediate containing a duplex and ssDNA moiety, BRCA2_{T2} would bind to the dsDNA close to the ssDNA giving BRCA2 the specificity for dsDNA/ssDNA junctions. This specificity might be 'diluted' in the context of the full-length protein in vitro as the CTD

binds more readily to ssDNA and doesn't bind dsDNA, however, it could be different in vivo. In any case, we think that the two-fold increase in activity on the full-length protein previously reported should come from the NTD.

2) Page 5, line 106: It is unclear, what the wording 'artificial' means in this context. We mean "artificial" as opposed to "natural variants" found in breast cancer patients as it is the case for C315S and C341S. We have now removed this term and state that the third mutation is a substitution of C279 to alanine.

3) Page 6, line 128: Please give the optimal concentrations of RAD51 and RPA used in the experiments.

We have now included the concentration used in the methods and the main text.

Reviewer #2 (Remarks to the Author):

The paper is describing the existence of a DNA binding domain in the N-terminus of BRCA2, which has not previously been identified. Further, the DNA binding characteristics of this domain show preference for dsDNA and ds/ss DNA interfaces. There is contrast with the DNA binding domain of the C-terminal domain of BRCA2, which shows more ssDNA and 3'-overhang preference. Putting this new information together, it suggests that BRCA2 acts as a mediator for RAD51 by binding both dsDNA and ssDNA either side of a transition to allow RAD51 nucleation.

The evidence presented looks to be unveiling an interesting new twist to understanding the function of BRCA2. It should be noted that the biochemistry of the strand exchange reaction requires 10-fold more BRCA2-NTD fragment (T2 = aa250-500). I think this difference in stoichiometry is probably not unexpected, but should be discussed in the description of the model.

We thank the reviewer for this comment. This is absolutely correct. Indeed, the fact that the stimulation of DNA strand exchange observed with the NTD is lower than with the full-length protein is not surprising; first, the NTD alone binds with lower affinity to DNA than full-length BRCA2 (see new Extended Data Fig. 4b, c); second, the NTD is missing the DNA binding activity and stimulatory action of the CTD which is present in the full-length protein. We have now included a paragraph in page 6 (lanes 150-155) where we acknowledge this difference.

The importance of the cysteines in this T2 fragment is emphasized by the effect of mutating one or two of the 3 cysteines in this fragment. In particular, the C315S mutant removes a significant amount of the activity.

Overall, there appears to be novel findings that will be of interest to the BRCA2

community.

Reviewers' Comments:

Reviewer #1 (Remarks to the Author)

Aura Carreira, Catharina von Nicolai, Åsa Ehlén, Taha Shahid, Yueru Sun, Charlotte Martin, and Xiaodong Zhang

The revised version of the manuscript is significantly improved and the authors have added new data that strengthen the case, eliminated the EM data, which did not add much information, and clarified the presentation in several cases. There are a few remaining issues that need to be addressed:

1) Several figures were assembled without attention to detail. Figure 1a has issues, so has Figure 2c. The journal may provide some advice on how to improve the visual presentation of the figures.

2) Figure S4A: The quality of the BRCA2 protein preparation is low, but given the size of the protein, this is a very vexing problem. The authors should state whether the additional bands are contamination or degradation products. Since the bands comprise a majority of the preparation this affects the interpretation of binding data.

3) Figure S4B: The graph shows a data point for 1250 nM for BRCA2 T2, which is not present on the gel. Is this a mistake or are the data on other gels? In the latter case, please show a gel that comprises the full range.

3) Figure S4C: The graph shows a data point for 250 nM for BRCA2 T2, which is not present on the gel. Is this a mistake or are the data on other gels? In the latter case, please show a gel that comprises the full range.

Reviewers' comments:

Reviewer #1 (Remarks to the Author):

Aura Carreira, Catharina von Nicolai, Åsa Ehlén, Taha Shahid, Yueru Sun, Charlotte Martin, and Xiaodong Zhang

The revised version of the manuscript is significantly improved and the authors have added new data that strengthen the case, eliminated the EM data, which did not add much information, and clarified the presentation in several cases. There are a few remaining issues that need to be addressed:

1) Several figures were assembled without attention to detail. Figure 1a has issues, so has Figure 2c. The journal may provide some advice on how to improve the visual presentation of the figures.

We thank the reviewer for this comment. We have aligned the objects and make the symbols consistent with other figures. We hope the outcome is satisfactory and we would be happy to get advice from the journal if needed for clarity.

2) Figure S4A: The quality of the BRCA2 protein preparation is low, but given the size of the protein, this is a very vexing problem. The authors should state whether the additional bands are contamination or degradation products. Since the bands comprise a majority of the preparation this affects the interpretation of binding data.

We thank the reviewer for this important remark. As mentioned in reviewer comment #2, purifying BRCA2 is a daunting task not only because of its big size (~460KDa including the tags) but because of its instability giving yields of few micrograms for each preparation. That said, we have now included a new gel and WB in Extended Data Fig. 4a that represents better the quality of our purified protein.

We have not re-verified by mass spectrometry the nature of the contaminants in the preparation used in this manuscript but we assume they would be very similar to the ones we have reported before (Jensen *et al.*, 2010; Martinez *et al.*, 2016). The fact that a single band is detected in the Western Blot suggests that the bands observed in the gel correspond to contaminant proteins rather than to truncated or degraded BRCA2 (the epitope of the antibody maps to the central region of the sequence of BRCA2). Importantly, the concentration of the protein is always adjusted to account for the presence of contaminants and/or truncated BRCA2 for each preparation by measure of subtracting the contribution of detectable bands other than full-length BRCA2 (as it is now stated in the Methods section, line 247-249). This allows the DNA binding experiment (in particular in the one referred to by the reviewer shown in Extended Figure 4b and c) be quantified based on the real concentration of full-length BRCA2 protein. The concentration is also adjusted in all the other purified proteins and fragments used in this manuscript. Thus, even if the purification of full-length BRCA2 might not be ideal, we are sure that the quantifications in the experiments shown in the manuscript are valid.

3) Figure S4B: The graph shows a data point for 1250 nM for BRCA2 T2, which is not present on the gel. Is this a mistake or are the data on other gels? In the latter case, please show a gel that comprises the full range.

This concentration was used in other gels but they do not have a gel with the whole range as we are using 10 well gels and comparing BRCA2 and BRCA2_{T2}

run in the same gel side by side. To avoid confusion, we have now removed this data point from the graph in Extended Data Fig. 4b and Fig. 2b.

3) Figure S4C: The graph shows a data point for 250 nM for BRCA2 T2, which is not present on the gel. Is this a mistake or are the data on other gels? In the latter case, please show a gel that comprises the full range.

This concentration was included in some of the gels, in particular those shown in Figure 2. To avoid confusion we have removed this data point from the graph in Extended Data Figure 4c as suggested by the reviewer.

Reviewer #1 (Remarks to the Author)

The authors made the requested changes.